# Induction of Hair Keratins Expression by an Annurca Apple-Based Nutraceutical Formulation in Human Follicular Cells

**DOI:** 10.3390/nu11123041

**Published:** 2019-12-13

**Authors:** Marialuisa Piccolo, Maria Grazia Ferraro, Francesco Maione, Maria Maisto, Mariano Stornaiuolo, Gian Carlo Tenore, Rita Santamaria, Carlo Irace, Ettore Novellino

**Affiliations:** Department of Pharmacy, School of Medicine and Surgery, University of Naples “Federico II”, Via D. Montesano 49, 80131 Naples, Italy

**Keywords:** keratins, primary human skin models, protein expression, nutraceuticals, Annurca apple, Procyanidin B2, antioxidants, hair growth

## Abstract

Hair disorders may considerably impact the social and psychological well-being of an individual. Recent advances in the understanding the biology of hair have encouraged the research and development of novel and safer natural hair growth agents. In this context, we have previously demonstrated—at both preclinical and clinical level—that an Annurca apple-based dietary supplement (AMS), acting as a nutraceutical, is endowed with an intense hair-inductive activity (trichogenicity), at once increasing hair tropism and keratin content. Herein, in the framework of preclinical investigations, new experiments in primary human models of follicular keratinocytes and dermal papilla cells have been performed to give an insight around AMS biological effects on specific hair keratins expression. As well as confirming the biocompatibility and the antioxidant proprieties of our nutraceutical formulation, we have proven an engagement of trichokeratins production underlying its biological effects on human follicular cells. Annurca apples are particularly rich in oligomeric procyanidins, natural polyphenols belonging to the broader class of bioflavonoids believed to exert many beneficial health effects. To our knowledge, none of the current available remedies for hair loss has hitherto shown to stimulate the production of hair keratins so clearly.

## 1. Introduction

Throughout the keratinization process and hair fibers production, numerous keratins organize into protein filaments to participate in the assembly of the hair shaft within follicle bulbs [1,2,3]. The hair fibers basically contain α-keratin and amorphous keratin, whose molecular structure is not yet very clear [4]. Similarly, the keratinization process—during which keratin is formed and its structure becomes organized and stabilized—has not been completely elucidated [5]. Among the different families and subfamilies of intermediate filaments (IF) proteins—representing the chemically stable basement of the most eukaryotic cytoskeletal systems—keratins are outstanding due to high molecular and functional diversity [6,7,8]. They are the typical intermediate filament-forming proteins of epithelial cells, providing mechanical support and a variety of essential biological functions at both cellular and tissue level by forming a dense and stable network spanning from the cell periphery to the nucleus of keratinocytes [9]. In addition, keratins participate in the regulation of cell growth, proliferation, migration and apoptosis [7]. Within a large multigene family evolved over the range of millions of years, in humans 54 functional keratin genes are known, i.e., 28 type I (acidic) and 26 type II (basic to neutral) keratins, expressed in highly specific patterns based on epithelial cell and appendage type, as well as on cellular differentiation, thus making up two of the largest sequence homology groups of heteropolymeric filaments [10].

In the last decades many hair follicle-specific epithelial keratins were discovered, which are built up from a somewhat separate gene subfamily [11,12]. They differ from the canonical epithelial keratins by considerably higher sulphur content in their non-α-helical head and tail domains, which is mainly responsible for the extraordinary high degree of filamentous cross-linking by keratin-associated proteins (KAPs) [13]. Indeed, the 28 type I genes comprise 17 genes encoding type I epithelial keratins (“soft” or “cyto-”) and 11 genes coding for type I hair keratins (“hard” or trichocytic”). Out of the 26 type II genes, 20 are epithelial keratin genes and only six encode hair keratins. However, as many as 26 of the total 54 epithelial keratins are specifically expressed in the hair follicle, and the total number of hair follicle-specific keratins roughly equals that of those expressed in other epidermis districts. Given their biological functions, mutations in most of them are associated with specific tissue fragility disorders as well as hair diseases [11,14]; as well, antibodies to keratins are important markers of tissue differentiation and tools in diagnostic pathology [15].

Hair keratins constitute up to 95% of the hair structure, and they are synthesized by specialized hair follicular keratinocytes, stimulated within the hair bulb by mediators released from follicular cells of dermal origin. Elongation and keratinization processes occur without loss of hair follicle morphology and proceed through proper control and organization of the complex hair keratin expression pattern [16]. This process is highly regulated and can be altered by a series of genetic, hormonal, and environmental factors, including possible nutrient deficiencies due to incorrect and unbalanced diets [17].

In this context, several pharmaceutical products have been formulated over the past decades for the treatment of alopecia and pattern baldness, but relatively minimal data on their efficacy and toxicity have been published [18,19]. In all its forms, baldness affects about 50% of the adult population, both male and female, and it represents a serious psychological distress in human society. Androgenetic alopecia (AGA), the most common form of hair loss in men and women, is a non-stop process which flows toward a definite pattern in genetically predisposed individuals [20]. Typically, the initial stages in all the main forms of baldness are linked by the presence of very thin hair fiber that easily forms coils within the hair follicle, indicating a defect in hair keratinization. In addition, genetic or acquired disorders prevent generation of well differentiated hair follicles, which is related to abnormal hair fiber keratinization [21]. To our knowledge, none of the available remedies have shown to directly affect the production of keratins, as well as the keratinization process. In consequence, the biomedical attention has focused on the discovery of new and safer natural products, such as oligomeric procyanidin able to promote hair epithelial cell growth and induce the anagen phase [22,23].

Very recently, we have demonstrated at both preclinical and clinical level that an Annurca apple-based dietary supplement (AMS), acting as a nutraceutical, effectively promotes hair growth and tropism, at once increasing hair density, weight, and primarily keratin content. Oral intake of 800 mg in two administrations a day of AMS for two months produces significant effects at the level of the hair follicles [24]. *Malus pumila Miller* cv. Annurca is a widespread apple accounting for 5% of Italian apple production, listed as a Protected Geographical Indication (PGI) product from the European Council (Commission Regulation (EC) No. 417/2006)). Procyanidin B2, a dimeric derivative present just in large amounts in the Annurca apple, has proved to be one of the most effective natural active ingredients in promoting hair growth, both in vitro and in humans by topical applications [25,26]. Indeed, AMS specifically stimulates the expression and biosynthesis of cytokeratins in a human epidermis model (HaCaT cells), without significant interference with the regulation of dynamic cellular processes such as growth and proliferation [24].

Here, in the framework of preclinical investigations to deepen understanding of the AMS biological effects on hair keratins synthesis, new experiments in primary models in vitro of human skin have been performed. Among human cells of the “hair cell system,” epidermal follicular keratinocytes and dermal papilla cells were ad hoc selected, as they are directly involved in hair biogenesis in vivo. Indeed, preclinical models give support to therapeutic efficacy testing and claim substantiation of hair growth modulators. On this path we have investigated the effects of AMS on the expression of specific keratins engaged in the production of hair. Based on the high number of polyphenols it contains, targeted bioscreens have confirmed the protective action toward oxidative stress and provided additional information on the safety of its use. In parallel, to give an insight into the mechanisms involved in keratins up-regulation, a keratinocytes/fibroblasts primary co-culture model of human skin, in part reproducing cell-cell interaction in the hair bulb system, were appropriately set up to explore protein biosynthesis following exposure to the nutraceutical formulation.

## 2. Materials and Methods

### 2.1. Apple Collection

Annurca (*M. pumila Miller* cv Annurca) apple fruits (each about 100 g) were collected in Valle di Maddaloni (Caserta, Italy) in October when fruits had just been harvested (green peel). The fruits were reddened following the typical treatment for about 30 days, and then analyzed.

### 2.2. Preparation of Annurca-Based Nutraceutical Formulation (AnnurMetS, AMS)

Annurca apples have been repeatedly extracted with water, and the obtained solution was filtered, centrifuged, and concentrated. Then, the solution was spry-dried in combination with maltodextrins, obtaining a fine powder named AnnurMetS (AMS) [24], actually branded as AnnurtriComplex^®^. The supplement was formulated by the Department of Pharmacy, University of Naples “Federico II” (Naples, Italy), while large-scale production was accomplished by MB-Med Company (Turin, Italy).

### 2.3. Primary Human Follicular Cells

To study AMS effects in preclinical trials, experiments were planned in specific cell models in vitro based on the use of primary human cells, ad hoc selected from the “hair cell system” and “skin cell system” categories. Epidermal follicular keratinocytes (human hair follicular keratinocytes, HHFK) and follicular fibroblasts belonging to the dermal papillae (human hair dermal papilla cells, HHDPC) were selected as cells directly involved in hair biogenesis, and purchased from ScienCellTM Research Laboratories. Cultures were established by means of specific dissection and dissociation protocols following surgical procedures (human scalp biopsies, ScienCell^®^ # 2440, TAN Record #944; and #2400 for HHFK and HHDPC, respectively) by appropriate donors based on specific phenotypic requirements (Caucasian race, male gender, age 55 years). Both HHFK and HHDPC were acquired at only one passage number and used immediately for the preclinical test. All the experiments herein reported were carried out with the cells at no more than 3–4 passages in vitro. HHFK cells were grown in Keratinocyte Medium (KM, ScienCell^®^) supplemented with 1% of keratinocyte growth supplement (KGS, ScienCell^®^) and 1% of penicillin/streptomycin solution (P/S, ScienCell^®^). HHDPC cells were grown in Mesenchymal Stem Cell Medium (MSCM, ScienCell^®^) supplemented with 5% of fetal bovine serum (FBS, ScienCell^®^), 1% of mesenchymal stem cell growth supplement (MSCGS, ScienCell^®^) and 1% of penicillin/streptomycin solution (P/S, ScienCell^®^). Both cell lines were cultured in a humidified 5% carbon dioxide atmosphere at 37 °C, according to ScienCell recommendations.

### 2.4. Bioscreens In Vitro

Targeted bioscreens were performed to define specific protocols for suitable treatments in vitro. Biochemical investigations were thereby performed to study AMS bioactivity in human hair follicular epidermal keratinocytes (HHFK cells) and in hair dermal papilla cells (HHDPC) in vitro, focusing on the evaluation of cell viability as well as on cell growth and proliferation following exposure up to 96 h to a range of concentrations of the nutraceutical formulation (0.5 ÷ 3 mg/mL). The range of in vitro concentrations used for preclinical evaluations was chosen on the basis of preliminary investigations [24]. In addition, by using high in vitro concentrations (up to 3 mg/mL in the bioscreen), we have tested the nutraceutical for undesirable cellular responses, to validate its safety, and for complete biocompatibility. In detail, the experimental procedure involved the estimation of a “cell survival index,” arising from the combination of cell viability evaluation with cell counting [27]. The cell survival index is calculated as the arithmetic mean between the percentage values derived from the MTT assay and the automated cell count, providing a more consistent indicator of cellular in vitro responses. To this aim, HHFK and HHDPC were inoculated in 96-microwell culture plates at a density of 10^4^ cells/well and allowed to grow for 24 h. The medium was then replaced with fresh medium and cells were treated for 24, 48, 72, and 96 h with different concentrations of AMS. Cell viability was evaluated using the MTT assay procedure, which measures the level of mitochondrial dehydrogenase activity using the yellow 3-(4,5-dimethyl-2-thiazolyl)-2,5-diphenyl-2H-tetrazolium bromide (MTT; Sigma, St. Louis, MO, USA) as substrate. The assay is based on the redox ability of living mitochondria to convert dissolved MTT into insoluble purple formazan. Briefly, after the treatments the medium was removed and the cells were incubated with 20 μL/well of MTT solution (5 mg/mL) for 1 h in a humidified 5% CO_2_ incubator at 37 °C. The incubation was stopped by removing the MTT solution and by adding 100 μL/well of DMSO to solubilize the obtained formazan. Finally, the absorbance was monitored at 550 nm using a microplate reader (iMark microplate reader; Bio-Rad, Hercules, CA, USA) [28]. Cell number was determined by the TC20 automated cell counter (Bio-Rad), providing an accurate and reproducible total count of cells and a live/dead ratio in one step by a specific dye (trypan blue) exclusion assay. Bio-Rad’s TC20 automated cell counter uses disposable slides, TC20 trypan blue dye (0.49 g/100 mL trypan blue dye in 0.81 g/100 mL sodium chloride and 0.06 g/100 mL potassium phosphate dibasic solution), and a CCD camera to count cells based on the analyses of captured images. Once the loaded slide is inserted into the slide port, the TC20 automatically focuses on the cells, detects the presence of trypan blue dye, and provides the count. When cells are damaged or dead, trypan blue can enter the cell allowing living cells to be counted. Operationally, after treatments in 96-microwell culture plates, the medium was removed and the cells were collected. Ten microliters of cell suspension, mixed with 0.4 g/100 mL trypan blue solution at 1:1 ratio, was loaded into the chambers of disposable slides. The results are displayed as total cell count (number of cells per milliliter). If trypan blue is detected, the instrument also accounts for the dilution and shows live cell count and percent viability. Total counts and live/dead ratio from random samples for each cell line were subjected to comparisons with manual hemocytometers in control experiments [29].

### 2.5. Cell Co-Culture System (HHFK and HHDPC)

Co-cultures have been prepared to study the effects of interactions between different types of cells by direct contact (juxtacrine) or diffusion of soluble factors (paracrine) in determining cellular responses to xenobiotics. A two-dimensional (2D) direct co-culture method in cell culture plates was used. Briefly, after trypsinization, HHFK cells were mixed in suspension with HHDPC cells at a 1:1 cell density ratio. The mixed cells (1 × 10^6^ cells) were then seeded in a petri culture dish (100 × 20 mm, Falcon^®^) and cultured in the respective media (described above) mixed in a 1:1 ratio. After 48 h of growth in a sub confluent culture, actual co-cultures (verified by microscopy) were exposed to specific concentrations of AMS (0.5 and 1 mg/mL) for 24, 48, 72, and 96 h. At the end points, bioscreens for cell viability and keratins expression analysis were performed.

### 2.6. Cytomorphological Analysis

HHFK and HHDPC primary cells were grown on 60 mm culture dishes by plating 5 × 10^5^ cells. After reaching the sub confluence, cells were incubated for 48 h with concentrations of AMS under the same in vitro experimental conditions described above and were then morphologically examined by a phase-contrast microscope (Labovert microscope, Leizt). Microphotographs at a 200× total magnification (20× objective and 10× eyepiece) were taken with a standard VCR camera (Nikon, Minato, Tokyo, Japan).

### 2.7. Oxidative Stress Induction and ROS Detection In Vitro

In order to study the antioxidant capacity of AMS in HHFK cells, intracellular reactive oxygen species (ROS) were detected by the ROS assay kit based on the use of 2′,7′-dichlorodihydrofluorescein diacetate (DCFDA Cellular ROS Detection Assay Kit, Abcam, Cambridge, UK). DCFDA is a fluorescent probe that can detect reactive oxygen species, e.g., hydrogen peroxide, peroxyl, and hydroxyl radicals [30]. After diffusion into the cell, DCFDA is deacetylated by cellular esterases to the non-fluorescent DCF product which cannot diffuse out of cells. DCF is then selectively oxidized by ROS in the fluorescent 2′,7′ dichlorofluorescein which can be detected using fluorescence spectroscopy at 485 nm excitation and 535 nm emission wavelengths. Briefly, HHFK cells were seeded into 96-well plates at a cell density of 2.5 × 10^4^ cells (× well). After 24 h of incubation at 37 °C in a 5% CO_2_ humidified atmosphere, cell monolayers were incubated for 48 h with or without AMS (0.5 and 1 mg/mL), and for an additional 24 h with the ion source of iron ferric ammonium citrate (FAC, 50 and 100 µg/mL) [31]. After treatments, cells were washed and incubated for 45 min at 37 °C in the dark with 25 μM DCFDA diluted in 1 × buffer according to the manufacturer’s instructions. Endpoints from triplicate wells for each experimental condition were measured in a fluorescence microplate reader (GloMax Explorer, Promega, Madison, WI, USA) at 485 nm excitation and 535 nm emission wavelengths.

### 2.8. Preparation of Cellular Extracts

HHFK, HHDPC, and co-cultures HHFK + HHDPC (1:1) were grown for 48 and 72 h in the presence or absence of 0.5 and 1 mg/mL of AMS nutraceutical formulation as previously described. Following incubations, cells were collected by trypsinization and pellets were lysed at 4 °C for 10 min with a buffer containing 1 g/100 mL Triton X-100, 5 mM EDTA, in PBS (pH 7.4) containing protease inhibitors. After centrifugation at 14,000× *g* for 10 min at 4 °C, the supernatant was collected as the soluble fraction and stored at −80 °C [32]. Protein concentration was determined by the Bio-Rad protein assay (Bio-Rad, Milan, Italy).

### 2.9. Western Blot Analysis

Keratins expression studies were performed at 0.5 ÷ 1 mg/mL concentrations of AMS in vitro. After treatments, defined amounts (50 μg) of proteins from cellular extracts were loaded and separated on a 10% SDS-PAGE and electro transferred onto a nitrocellulose membrane (0.2 µm nitrocellulose membrane, Trans-Blot^®^ Turbo^TM^, Transfer Pack, Bio-Rad Laboratories, Hercules, CA, USA) using a Bio-Rad Transblot (Bio-Rad). Proteins were visualized by reversible staining with Ponceau-S solution and destained in PBS, and membranes were blocked at room temperature in a milk buffer (1 × PBS, 5–10 g/100 mL nonfat dry milk, 0.2% g/100 mL Tween-20). For cytokeratins immunodetection, membranes were incubated at 4 °C overnight with 1:250 monoclonal anti-pan cytokeratin antibodies (mixture) (Sigma-Aldrich, Milan, Italy), which recognize the following human cytokeratins (“soft-keratins”), according to their molecular weight: K1 (68 kDa), K4 (59 kDa), K5 (58 kDa), K6/K10 (56 kDa), K13 (54 kDa), K8 (52 kDa), K18 (45 kDa), and K19 (40 kDa). For hair keratins analysis, membranes were incubated with 1:1000 polyclonal anti-Type I + II Hair Keratins (human) guinea pig polyclonal antibody (Progen, Cat. No. GP-panHK), which recognizes synthetic peptides common to human type I (acidic) and type II (basic) hair (“trichocytic” or “hard”) keratins: K31–K40 (former designation hHa1-hHa8) and K81–K86 (former designation hHb1–hHb4). A goat anti-mouse IgG + IgM (1:5000; Jackson ImmunoResearch Laboratories, Baltimore Pike, West Grove, PA, USA) and a rabbit anti-guinea pig IgG (H + L) (1:4000; ThermoFisher, Cat. No. 61-4620) were used as secondary antibodies, respectively. The immunocomplexes were visualized by the ECL chemiluminescence method (Clarity^TM^ Western ECL Substrate, Bio-Rad Laboratories, Hercules, CA, USA) and analyzed by an imaging system (ChemiDoc, Imaging System, Bio-Rad). The GAPDH antibody (Sigma-Aldrich) was used as housekeeping gene to normalize the results [31].

### 2.10. Statistical Analysis

All data were presented as mean values ± SEM of *n* experiments. The statistical analysis was performed using a Graph-Pad Prism (Graph-Pad software Inc., San Diego, CA, USA), and an ANOVA test for multiple comparisons was performed followed by Bonferroni’s test.

## 3. Results

### 3.1. Cellular Response to AMS Nutraceutical Formulation In Vitro

In continuity with preclinical studies performed on HaCaT cells [24], selected bioscreens on human hair follicular keratinocytes (HHFK) and human hair dermal papilla cells (HHDPC) were performed in order to evaluate cell responses to AMS treatment. For this purpose, experimental protocols in vitro reproducing systemic administration of AMS in vivo were accomplished. The experimental design was based on tests with AMS concentrations ranging from 0.1 to 3 mg/mL, in the context of time course experiments between 24 and 96 h. The obtained data—reported as “Cell survival index” in concentration-effect curves (Figure 1)—plainly provide evidence of no significant interference with cell viability, even at the highest concentration (3 mg/mL) and after long times of incubation in vitro (96 h), thus giving additional evidence to guarantee safety in the use of this nutraceutical. Rather, bioscreens show a mild induction of cell growth and proliferation in both treated keratinocytes and dermal papilla cells with respect to untreated control cultures. However, the increase in viability and proliferation of both pure cell lines becomes significant only after 48 h of treatment and with AMS concentrations in vitro equal to or less than 1 mg/mL. Therefore, exclusively low concentrations of AMS can weakly stimulate the proliferation of these cellular systems. Further specific bioscreens in vitro on a primary co-culture model of HHFK and HHDPC have excluded any cytotoxic effects of AMS nutraceutical formulation. Moreover, AMS-stimulating effects on hair follicular cells are more evident when they are co-cultured in appropriate experimental conditions. For support, we monitored cellular morphology throughout the preclinical test. Endpoints imaging by phase-contrast light microscopy were taken from cells to assess prospective cellular monolayer modifications induced by AMS treatment. Microphotographs in Figure 2 endorse the biocompatibility of AMS on subconfluent monolayers of follicular keratinocyte and dermal papilla cells. Based on these results, to avoid using high concentrations in vitro, which would also be impossible to reach in vivo at the hair follicles after systemic administration of AMS supplement, we chose to continue the study at the lowest concentrations of 0.5 and 1 mg/mL.

### 3.2. Antioxidant Protection in AMS-Treated Hair Follicle Cells

To evaluate the occurrence of protective antioxidant effects, HHFK and HHDPC treated for 48 h with 0.5 and 1 mg/mL of AMS formulation were subsequently exposed, based on a well-established protocol, to high concentrations of FAC as an ion source of iron. In the presence of oxygen and normal cellular metabolic activities, cellular iron overload rapidly generates oxidative stress by increasing ROS, with a substantial impairment of cell viability due to oxidative injury [33]. In both cell types, the presence of AMS allows to significantly control the induction of oxidative stress and to concurrently limit cellular oxidative damage thereby preserving viability of hair follicular cells (Figure 3). Approximately, the trend in both the generation of oxidative stress and cellular responses to treatments is comparable. However, data depicted in Figure 3 are suggestive of an increased susceptibility to oxidative stress of follicular keratinocytes (Figure 3a), which are properly protected under these experimental conditions by the presence of the AMS molecular components.

### 3.3. AMS Biological Effects on Hair Keratins Expression in HHFK

To explore AMS biological effects on the regulation of hair keratin expression, hair follicular keratinocytes were cultured in standard growth medium in the presence of 0.5 and 1 mg/mL of the nutraceutical formulations for 48 and 72 h, as determined by preliminary bioscreens. Regarding the analysis of protein expression, after incubations the total cellular levels of keratins were evaluated by immunoblot analysis using a special antibody mix (broad-spectrum anti-pan) selectively directed towards both types I and II keratins present in the hair (anti-Type I + II Hair Keratins (human) guinea pig polyclonal, serum). Following this path, we have obtained detailed information on type I (K31–K40) and type II (K81–K86) hair keratins expressed by hair follicular keratinocytes. Indeed, an overall keratins analysis concerning the expression profile and protein pattern by Western blot showed a significant concentration- and time-dependent increase in hair keratin isoforms after cultures exposure to AMS (Figure 4a). Starting from 48 h of incubation, and at the concentrations of 0.5 and 1 mg/mL, AMS induced a considerable increase in cellular content of almost all detectable hair keratins (up to 300% of average keratin content following 72 h of treatment compared to untreated keratinocytes, Figure 4b). In more detail, among type I hair keratins, the most up-regulated were K31, K35, and K36 after 72 h of treatment in vitro, whilst K82 and K85 were the most up-regulated for type II hair keratins. Regarding K81, K83, K85, and K86 (all types II hair keratin isoforms), increased cellular content was already detectable after 48 h of treatment. Finally, enhanced levels of K32 and K33a/b were quantifiable exclusively after 48 h of incubation. Overall, this outcome demonstrates an important AMS-dependent up-regulation in the production of hair keratins via a direct stimulus on HHFK.

### 3.4. AMS Effects on Cytokeratins Expression in HHDPC

After treatments in vitro of dermal papilla cells with AMS formulation by the same experimental procedure, the total cellular levels of cytokeratins were evaluated by western blot analysis using a broad-spectrum anti-pan-cytokeratin antibody, as described in the experimental section. AMS determined a significant increase in the expression of cytokeratins also in non-epidermal cells of the hair bulb. As in the case of HHFK, this effect is concentration- and time-dependent, reaching important increase values in total cytokeratin content with respect to untreated dermal papilla cells (Figure 5). It can be concluded that AMS is able to enhance the keratins production in both epidermal and non-epidermal cells at the hair follicle level.

### 3.5. Effect of AMS Formulation on Hair Keratins Expression in a Co-Culture Model of HHFK and HHDPC

In order to partially reproduce the cellular interactions stimulating hair keratins expression and the growth of skin production at the hair bulb level, a cellular co-culture model was set up starting from primary cells of follicular keratinocytes (HHFK) and dermal papilla cells (HHDPC) as described in the experimental section. Dermal papilla cells are located at the base of hair follicles and play a key role in all regulatory processes that control hair growth and proliferation. Following the same experimental protocol, the co-cultures were incubated for different times (48 and 72 h) with 0.5 and 1 mg/mL of AMS. After the treatments in vitro, the cells were collected and suitably lysed to obtain total protein extracts. Subsequently, immunodetection assay was conducted by broad-spectrum anti-pan antibodies to investigate the cellular content of total hair keratins (Figure 6). The results show a further significant increase in protein production. The increase in hair keratins expression induced by AMS nutraceutical formulation was very consistent with respect to untreated co-cultures, but also compared to that observed in HHFK cultures alone, as underscored in the line graph of Figure 6b.

## 4. Discussion

Baldness and patterned hair loss affect around 50% of the adult population worldwide. Independent of age and gender, these conditions can produce a negative influence on life quality and a large psychological impact on patients. Their aetiology is complex and may involve a combination of genetic, metabolic, psychological, and environmental factors [34]. Typically, hair follicles undergo a premature miniaturization and shorten their permanence in anagen [20]. Abnormalities in keratinization lead to the thinning of the fibers, anticipating hair loss. In the case of androgenic alopecia, one of the most common cause of patterned hair loss, pathogenesis has been partially elucidated and proven to involve an altered metabolism of testosterone with an increased activity of the type II isoform of the 5-α reductase enzyme (5AR) [35]. Finasteride, the only United States Federal Drug administration (FDA)-approved oral agent for the treatment of hair loss, is a specific inhibitor of 5AR type II isoform, while Minoxidil, the other FDA-approved agent, acts topically as a potassium channel opener, increasing vascularization of the hair bulb [36,37]. However, considering the occurrence and the complexity of hair loss, as well as the adverse effects by available drugs, in recent years research for novel remedies and less dangerous therapies has been very active.

Nutraceuticals have been hitherto shown to be safe and effective treatment options, so that an increasing number of reports have focused on natural products among nutritional and antioxidant therapies as alternatives for hair loss treatment. Nonetheless, few of these have clinically proved to be effective [38]. In this context, recent preclinical evidence and clinical trials have shown an Annurca apple extract (AMS) as a nutraceutical able to effectively counteract adult baldness and patterned hair loss caused by various factors, including androgenetic and metabolic ones. Indeed, significant results in terms of increased hair growth, hair number, hair weight, and keratin content could be measured after two months of oral treatment [24]. The active ingredient behind this formulation is a dry extract derived from the Annurca apple, a variety particularly rich in oligomeric procyanidins, among natural polyphenols belonging to the broader class of bioflavonoids [39,40]. Procyanidin B2 has now proven to be effective in promoting hair growth and proliferation. In murine epithelial cells, Procyanidin B2 has been shown to inhibit protein kinase C (PKC) isozymes and modulate transforming growth factors β (TGF β), a class of cytokines whose signaling pathway plays critical roles in regulating different cellular activities including cell growth, cell differentiation, and cell development [23,25,26]. To our knowledge, beyond the biological effects of procyanidins, our findings have shown for the first time an important increase in keratin biosynthesis and content in human epidermal cells, which could play an important role in the growth of skin production by favoring the keratinization process [24].

To strengthen AMS biological effects in a preclinical test, experiments have been herein performed in specific cell models in vitro based on primary human cells obtained by the hair follicle. Compared to preliminary studies performed on the HaCaT cell line, hair follicle primary cultures directly originate by human biopsies from specific skin areas (typically occipital and temporal regions of the head). As well, the donors’ type (race, sex, age, pigmentation, etc.) was carefully selected in order to ensure the best experimental conditions to achieve targeted investigations. Among available cell models, epidermal follicular keratinocytes and follicular fibroblasts belonging to the dermal papillae were selected as directly involved in hair biogenesis [41]. Follicular keratinocytes are in fact implicated in the exclusive expression of keratins constituting the hair structure (K81–K86 (type II), K31–K40 (type I)), located both in the cortex and in the cuticle, representing the most important cells within the hair follicle involved in hair growth [11,12,13]. Dermal cells are specialized mesenchymal cells located at the base of the hair follicle that play an essential role in hair follicular morphogenesis and in all regulation processes controlling postnatal hair growth cycles [42]. Overall, by AMS treatment in vitro we found a remarkable increase of hair keratins expression in keratinocytes, as well as of cytokeratins in dermal papilla cells, already at concentrations of 0.5 and 1 mg/mL of the nutraceutical formulation. Consistent with current in vivo findings concerning total keratin content, the increase in hair keratins production becomes even more consistent when cells are grown in co-culture models. Keratinocytes are in fact very receptive to stimulatory factors produced by papilla cells regulating their proliferation as well as protein synthesis. Indeed, hair-inductive activity (trichogenicity) requires interactions between dermal and epiderma cells, allowing the maintenance of a delicate balance between cell proliferation and differentiation in active hair follicles [43]. In line, hair shaft of mice treated topically with an AMS-based cosmetic foam showed an increased content in sulfur which correlates to the enhanced content of keratins produced by the hair follicle in anagen [44]. In addition, topical treatment for four weeks significantly changed the levels of several key intracellular metabolites by redirection of hair bulb metabolism to a mitochondrial elevated activity and ATP production by mainly lipid β-oxidation. Accordingly, the hair follicle can likely save amino acids pool so that protein biosynthesis and hair keratins production were markedly stimulated by AMS treatment [44]. On balance, mitochondria have been already shown to contribute to hair bulb energy metabolism, and mitochondrial β-oxidation favorably affects hair growth in vitro and in vivo [45,46]. Molecular components of AMS give the impression of strengthening this metabolic setting, thus favoring protein synthesis. To support, keratinocytes mitochondrial breakdown decreases hair follicle density, increases apoptosis, and reduces proliferation [47]. Stimulation of mitochondrial function with thyroid hormones prolongs anagen and modifies intrafollicular keratin expression [48]. In addition, in the hair follicle pentose phosphate pathway rate and nucleotides production are reduced by AMS treatment [44]. In a context of total biocompatibility of the nutraceutical, as DNA replication and mitosis are not stimulated, we found only weak effects on proliferation induction both in pure hair follicle cultures and in co-culture systems, just detectable at very low concentrations (equal to or less than 1 mg/mL).

Finally, AMS redox protective effects are not to be overlooked. While the role of oxidative stress has been widely discussed in skin aging, little focus has been placed on its impact on hair condition [49]. The data presented are revealing of the AMS’ ability to protect follicular keratinocytes against oxidative stress. Looking for improved nutraceutical strategies able to warrant beneficial effects on hair growth and trophism, the occurrence of antioxidant properties could be adjuvant to support supplements bioactivity by preserving vitality and activity of follicular cells, even in the presence of numerous dietary, environmental, and metabolic cytotoxic factors. As well, proteins including “soft” keratins are important targets for oxidative modifications [50]. Oxygen radicals and other activated oxygen species generated as by-products of cellular metabolism or from environmental sources can trigger modifications that generally result in functional changes in structural proteins [51]. As protein oxidation can be associated with an increased susceptibility to proteases, the antioxidant potential of AMS can be an important player for the keratinization process [52]. Moreover, based primarily on the polyphenolic pool of the Annurca apple extract, it has recently been suggested that there is a biological interference with the WNT signaling, which deserves further investigations according to its critical role in hair cell specification and development [53].

Overall, the results of this study, together with the previously reported findings, suggest that consumption of AMS nutraceutical formulation stimulates the production of hair keratins and provides protection for the hair cell against oxidative damage.

## 5. Conclusions

*Malus pumila Miller* cv. Annurca is an apple native to Southern Italy particularly rich in oligomeric procyanidins, among natural polyphenols belonging to the broader class of bioflavonoids. As part of a multidisciplinary study to develop a nutraceutical supplement, our recent findings proved oral consumption of an Annurca apple extract is capable of an intense hair-inductive activity (trichogenicity) in healthy human subjects, at once increasing hair growth and tropism. By novel preclinical evaluations in primary models of skin, we have demonstrated for the first time an engagement of trichokeratins expression underlying the Annurca extract’s biological effects on human follicular cells.

## Figures and Tables

**Figure 1 nutrients-11-03041-f001:**
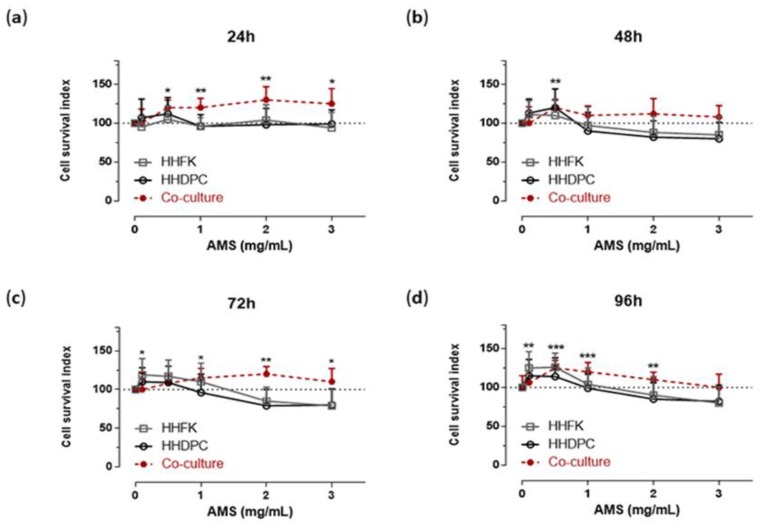
Bioscreen for cellular responses to Annurca apple-based dietary supplement (AMS) treatment in vitro. Cell survival index, evaluated by the MTT assay and live/dead cell ratio, for human hair follicular keratinocytes (HHFK), human hair dermal papilla cells (HHDPC) and HHFK-HHDPC co-cultures following 24 (**a**), 48 (**b**), 72 (**c**) and 96 h (**d**) of incubation with the indicated concentration (the range 0.5–3 mg/mL has been explored) of AMS supplement, as indicated in the legend. Results are expressed in line graphs as percentage of untreated control cells and are reported as mean of four independent experiments ± SEM (*n* = 20). *** *p* < 0.001 vs. control (untreated cells); ** *p* < 0.01 vs. control (untreated cells); * *p* < 0.05 vs. control (untreated cells).

**Figure 2 nutrients-11-03041-f002:**
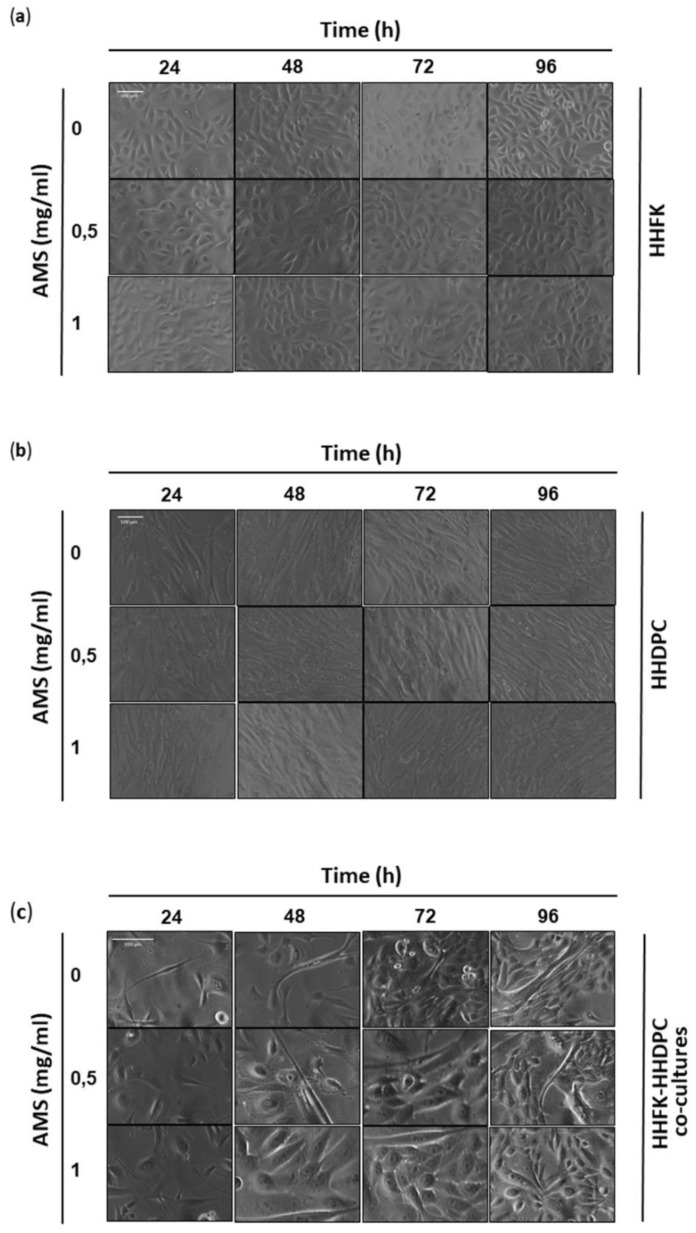
Cytomorphological analysis on cell monolayers. Representative microphotographs by phase-contrast light microscopy at a 100× magnification (10× objective and a 10× eyepiece) of HHFK (**a**) and HHDPC (**b**), and at a 200× magnification (20× objective and a 10× eyepiece) of HHFK-HHDPC co-cultures (**c**) treated for the indicated times (from 24 to 96 h) with 0.5 and 1 mg/mL of AMS nutraceutical, as indicated. The shown images are representative of three independent experiments.

**Figure 3 nutrients-11-03041-f003:**
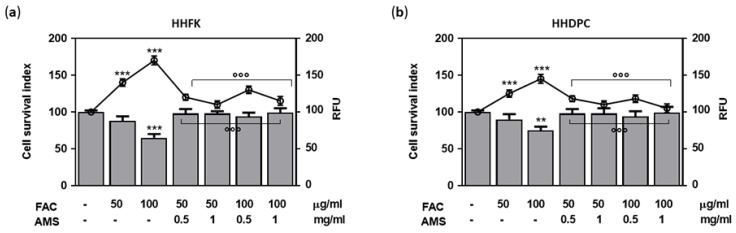
Antioxidant effects by AMS in hair follicle cells. HHFK (**a**) and HHDPC (**b**) were pre-incubated for 48 h with 0.5 and 1 mg/mL of AMS formulation and subsequently exposed for 24 h to different concentrations (50 and 100 µg/mL) of FAC (Ferric (III) Ammonium Citrate) to simulate cellular iron overload. Following treatments in vitro, cell viability was determined by the “Cell survival index,” whilst reactive oxygen species (ROS) detection was ensured fluorescently (RFU, Relative Fluorescence Units) by the selective oxidation of the non-fluorescent DCF probe in the fluorescent product 2′,7′ dichlorofluorescein. *** *p* < 0.001 vs. control (untreated cells); ** *p* < 0.01 vs. control (untreated cells); °°° *p* < 0.001 vs. 100 µg/mL FAC-treated cells.

**Figure 4 nutrients-11-03041-f004:**
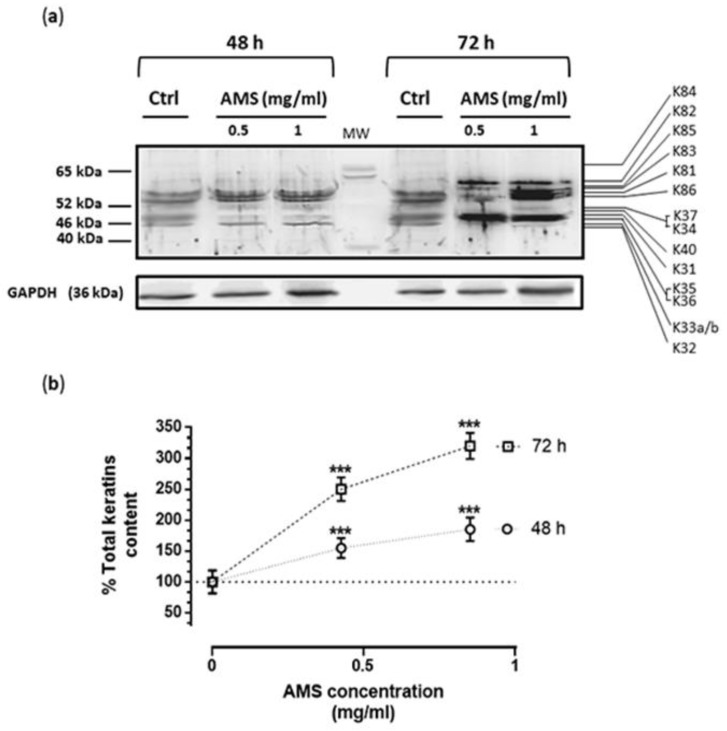
Induction of hair keratin expression in HHFK by AMS supplement. (**a**) Immunodetection analysis showing the total hair keratin levels in HHFK exposed or not (Ctrl) for 48 and 72 h to AMS supplement (0.5 and 1 mg/mL), as indicated in the figure. Membranes were incubated with 1:1000 polyclonal anti-Type I + II Hair Keratins guinea pig polyclonal antibody, which recognize human type I (acidic) and type II (basic) hair (“trichocytic” or “hard”) keratins K31–K40 (former designation hHa1–hHa8), and K81–K86 (former designation hHb1–hHb4), as reported in the experimental section. The blot is representative of four independent experiments. (**b**) Hair keratin bands within each lane were quantified by densitometric analysis and plotted in a line graph as percentage of total keratins with respect to control. The anti-GAPDH antibody was used to standardize the amounts of proteins in each lane. Shown are the average ± SEM values of four independent experiments. *** *p* < 0.001 vs. control (untreated cells).

**Figure 5 nutrients-11-03041-f005:**
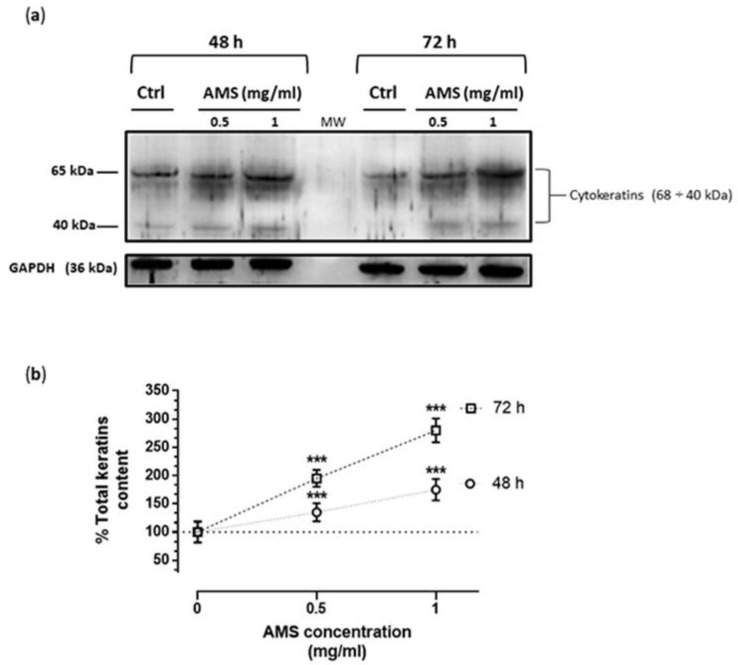
Cytokeratin content in HHDPC after incubation with AMS. (**a**) Following treatments in vitro of HHDPC by 0.5 and 1 mg/mL of AMS formulation for 48 and 72 h, equal amounts of proteins (50 µg) were separated on a 10% SDS–polyacrylamide gel and subjected to Western blot analysis using monoclonal anti-Pan cytokeratin antibodies, which recognize human cytokeratins (“soft keratins”), as specified in the experimental section. Ctrl represents untreated control cells. The blot is representative of four independent experiments. (**b**) The cytokeratin bands within each lane were quantified by densitometric analysis and plotted in line graph as percentage of total cytokeratins compared to untreated control. The anti-GAPDH antibody was used to standardize the amounts of proteins in each lane. Shown are the averages ± SEM values of four independent experiments. *** *p* < 0.001 vs. control (untreated cells).

**Figure 6 nutrients-11-03041-f006:**
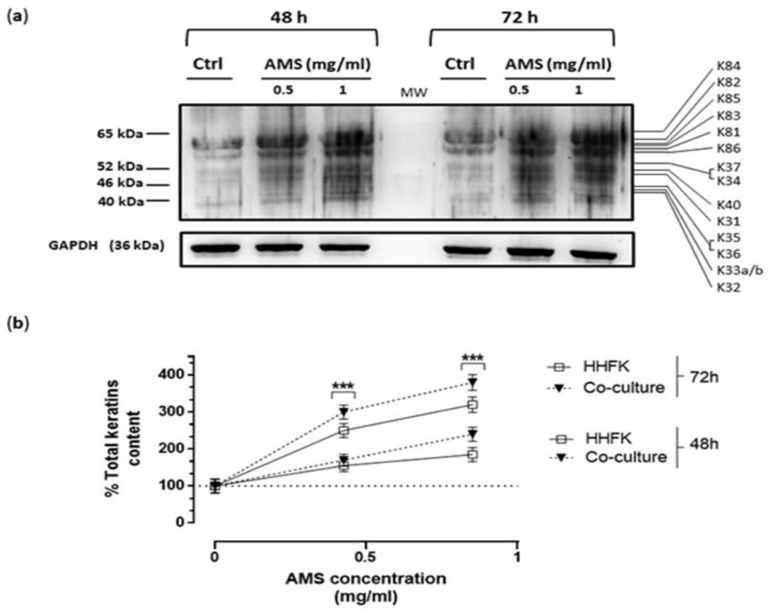
Enhanced hair keratins expression in a co-culture model of human hair cells. (**a**) HHFK and HHDPC co-cultures were cultured in presence or not (Ctrl) of AMS supplement (0.5 and 1 mg/mL) for 48 and 72 h. Then, cells were collected and lysed to obtain the protein fraction subjected to immunodetection analysis by using 1:1000 polyclonal anti-Type I + II human Hair Keratins polyclonal antibody. The shown membrane is representative of three independent experiments. (**b**) Total hair keratin content within each lane was quantified and plotted in a line graph as percentage of total keratins compared to untreated co-cultures. The anti-GAPDH antibody was used to standardize the amounts of proteins in each lane. Shown are the average ± SEM values of three independent experiments. *** *p* < 0.001 vs. control (untreated co-cultures).

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
