# Peer review of "Induction of Hair Keratins Expression by an Annurca Apple-Based Nutraceutical Formulation in Human Follicular Cells"

_nutrients, 2019, doi:10.3390/nu11123041_

Round 1

Reviewer 1 Report

In the work “Induction of Hair Keratins Expression by an Annurca Apple-Based Nutraceutical Formulation in Human Follicular Cells” the authors studied the potential of an Annurca Apple-Based Nutraceutical to induce the expression of trichokeratins. The authors used two primary human follicular cells, epidermal follicular keratinocytes (HHFK) and follicular fibroblasts belonging to the dermal papillae (HHDPC), to study the expression of hair keratins and cytokeratins, respectively. The results obtained support the role of the annurca apple-based nutraceutical as an enhancer of the trichokeratins expression, and support the great potential of this nutraceutical for the treatment of hair loss disorders.

The paper is well written, however, the abstract does not reflect the work developed and the results obtained. Please, improve the abstract considering this.

Authors should address some minor remarks:

Page 3 – “…range of concentrations of the nutraceutical formulation (0.5 ÷ 3 mg/mL).” – a justification for the selected range of concentrations should be included. Page 4 – “Cell viability was evaluated using the MTT assay procedure.” – please include a reference. Page 4 – “DCFDA is a fluorescent probe that can detect reactive oxygen species, e.g. hydrogen peroxide, peroxyl, and hydroxyl radicals.” – please include a reference In fig 1 it is evident a protective effect of the co-cultures for the highest concentrations of nutraceutical? An explanation of this protective effect should be included. Please include in fig 2 and 3 the type of cell used. Information was only included in the figure caption. In Fig 4, 5 and 6 captions, there is no information regarding the control.

Author Response

The manuscript has considerably benefited from the comments and suggestions of Reviewers. Therefore, we thank reviewers for their valuable advices. In what follows we reply to their suggestions.

Reviewer #1 comments:

In the work “Induction of Hair Keratins Expression by an Annurca Apple-Based Nutraceutical Formulation in Human Follicular Cells” the authors studied the potential of an Annurca Apple-Based Nutraceutical to induce the expression of trichokeratins. The authors used two primary human follicular cells, epidermal follicular keratinocytes (HHFK) and follicular fibroblasts belonging to the dermal papillae (HHDPC), to study the expression of hair keratins and cytokeratins, respectively. The results obtained support the role of the annurca apple-based nutraceutical as an enhancer of the trichokeratins expression, and support the great potential of this nutraceutical for the treatment of hair loss disorders.

1. The paper is well written, however, the abstract does not reflect the work developed and the results obtained. Please, improve the abstract considering this.

1. Based on the referee's suggestions, the abstract has now been improved.

2. Page 3 – “…range of concentrations of the nutraceutical formulation (0.5 ÷ 3 mg/mL).” – a justification for the selected range of concentrations should be included.

2. The range of concentrations used to perform bioscreen in vitro arises from a series of preliminary experiments aimed at establishing the effects of the nutraceutical formulation on human cells, both in terms of biocompatibility and safety, and in the biosynthesis of cytoskeletal proteins, such as keratins. Compared to the first report (Tenore et al., J Med Food 2018, 21 (1), 90-103) in which for the first time we discussed about the effects of AMS in preclinical models (by using a concentration range from 0.009 to 2.3 mg/ml for established times), the range of concentrations used herein is slightly shifted upwards. In this way, using very high in vitro concentrations (up to 3 mg/ml in the bioscreen), we have tested the nutraceutical for undesirable cellular responses, thus confirming its safety and complete biocompatibility. The expression studies of keratins were instead performed at lower concentrations in vitro in an attempt to get as close as possible to the actual concentrations reached in vivo at the site of action following oral administration of 800-1000 mg/die of the nutraceutical formulation. In the light of these considerations, we have included a sentence in the experimental section to justify this choice. Moreover, in the context of clinical trials and in models in vivo, new pharmacokinetic experiments are underway to define the blood levels as well as the concentrations at the site of action (hair follicles) of the active ingredients, based on the recommended dosage regimen of the nutraceutical AnnurtriComplex® (800 mg/die in two oral administration).

3. Page 4 – “Cell viability was evaluated using the MTT assay procedure.” – please include a reference.

3. A reference for the MTT assay procedure has been added.

4. Page 4 – “DCFDA is a fluorescent probe that can detect reactive oxygen species, e.g. hydrogen peroxide, peroxyl, and hydroxyl radicals.” – please include a reference.

4. Reference added.

5. In fig 1 it is evident a protective effect of the co-cultures for the highest concentrations of nutraceutical? An explanation of this protective effect should be included.

5. In the bioscreen shown in Figure 1 we cannot speak about protective effects of the AMS nutraceutical, simply because in the experimental conditions used no type of cellular stress and/or insult is used towards which the treatment with the nutraceutical should show cytoprotective effects. Rather, the effect  the referee has rightly highlighted is due to the fact that in co-culture systems and in appropriate experimental conditions (i.e. the presence of the nutraceutical), follicular keratinocytes and dermal papilla cells stimulate each other presenting overall values of viability higher than those of the single cells kept growing in pure cultures. This effect was however already discussed by us, reporting within the explanation of the results the sentence “AMS stimulating effects on hair follicular cells are more evident when they are co-cultured in appropriate experimental conditions”.

6. Please include in fig 2 and 3 the type of cell used. Information was only included in the figure caption.

6. The type of primary cells or co-cultures used in the various experiments depicted in Fig. 2 and 3 was now directly added in the figures.

7. In Fig 4, 5 and 6 captions, there is no information regarding the control.

7. Captions of Fig. 4, 5 and 6 have been improved by inserting information about control cells.

Reviewer 2 Report

The authors examined the effect of AMS on follicular keratinocytes and dermal papilla cells and demonstrated that AMS stimulated the production of hair keratins.

Major points

Although the authors described the methods in detail, the critical informations are lacking, for example; passage number of the cultured cells. They used one cell line each, HHFK and HHDPC, as primary human cells in this paper. It is very important to describe the passage number in case of primary cells because dermal papilla cells are known to lose proliferating potential and hair-inducing capacity after several passages. I don’t understand the reason why the authors used a “cell survival index” as a more consistent indicator of cellular in vitro responses, instead of the individual cell viability and cell counts data. I think those data should be presented as well as a cell survival index. The authors concluded the AMS stimulated the production of hair keratins “directly” based on the western blotting data. I wonder if the mere analysis of protein levels would be enough, because indirect effect cannot be ruled out after 48 or 72 hours. The message level analysis such as gene expression data would be helpful to say “directly”. I would like to know how much we need to take AMS orally to reach the blood concentration level more than 0.5mg/mL which seems to be effective in vivo.This information may be very valuable in this journal.

Minor points

There are many different words for describing dermal papilla cells in the text, such as follicular fibroblasts, dermal cells, dermal papilla, dermal follicular cells, papilla cells.I would like to recommend to unify as “dermal papilla cells” for avoiding confusion because this term is generally accepted in the hair research field. Please paragraph introduction and discussion to improve readability. Most of the abstract is occupied by the mere explanation of the background. Abstract needs to be rewritten because it does not express specific content. What is “RFU” described at the right side of the graphs in the Figure 3?

Author Response

The manuscript has considerably benefited from the comments and suggestions of Reviewers. Therefore, we thank reviewers for their valuable advices. In what follows we reply to their suggestions.

Reviewer #2 comments:

The authors examined the effect of AMS on follicular keratinocytes and dermal papilla cells and demonstrated that AMS stimulated the production of hair keratins.

Major points

1. Although the authors described the methods in detail, the critical informations are lacking, for example; passage number of the cultured cells. They used one cell line each, HHFK and HHDPC, as primary human cells in this paper. It is very important to describe the passage number in case of primary cells because dermal papilla cells are known to lose proliferating potential and hair-inducing capacity after several passages.

1.  We thank the Referee for his valuable suggestions and take the opportunity to improve technical information in the manuscript. Both follicular keratinocytes and dermal papilla cells were acquired by the cellular bank of ScienceCell at only one passage number and used immediately for preclinical test. Indeed, all the experiments herein reported were carried out with the cells at no more than 3-4 passages in vitro. This information has now been included in the experimental section (Primary human follicular cells).

2. I don’t understand the reason why the authors used a “cell survival index” as a more consistent indicator of cellular in vitro responses, instead of the individual cell viability and cell counts data. I think those data should be presented as well as a cell survival index.

2. The cell survival index, as exhaustively reported in the methods section, combines in a single value the automated cell count with a functional assay, i.e. the MTT assay based on the evaluation of mitochondrial redox activity. On the basis of our experience in the preclinical field for the development of new bioactive molecules (both drugs and nutraceuticals, for example Irace et al., Sci Rep 2017, 7, 45236 and Miniaci et al., J Cell Biochem. 2016, 117(2), 402-12) cell count alone could in some cases generate false positive results (by counting among viable cells those that have already started pathways of programmed death such as apoptosis and autophagy, and that soon they will no longer be viable). Thus, the combination of two different parameters gives rise to a more reliable indicator for analysing cellular responses in vitro. Deriving the cell survival index from the  combination of counts and MTT assays, we believe it is redundant to present both, also because these results, consistent with the previous ones, clearly demonstrate the safety and biocompatibility of this formulation without exceptions requiring further and more detailed investigation.

3. The authors concluded the AMS stimulated the production of hair keratins “directly” based on the western blotting data. I wonder if the mere analysis of protein levels would be enough, because indirect effect cannot be ruled out after 48 or 72 hours. The message level analysis such as gene expression data would be helpful to say “directly”.

3. It is true that there may be indirect effects influencing protein expression at the cellular level. Indeed, in order to give an insight concerning AMS-induced effects on different metabolic pathways, new microarray-based experiments are in progress to in depth analyse the effects of AMS nutraceutical formulation on a large number of genes at a transcriptomic level, including both soft and hard keratins. These will be the subject of a new manuscript in preparation. However, the immunoassay used in Western blotting procedure allows an overall assessment of the protein cellular content. Certainly, the data that we can understand by these experiments hang in favour of a consistent increase in the cellular level of keratins. Given that compared to untreated control cells we are speaking about an increase of hair keratins and cytokeratins content in the order of 300% and more (as for example in the case of co-cultures), it would be really surprising not to consider protein synthesis at the base of the observed effects. However, rather than to say “directly”, both in the results and discussion sections we intentionally state about the “increased protein content” and “total keratin content” as the actual data emerging from this experimentation. To this purpose and following referee’s suggestions, we have also revised the abstract.

4. I would like to know how much we need to take AMS orally to reach the blood concentration level more than 0.5mg/mL which seems to be effective in vivo. This information may be very valuable in this journal.

4. This is a very interesting point, but an exhaustive answer to this question requires several data that are not yet available. Indeed, new experiments are underway in preclinical models in vivo as well as in the context of clinical trials in order to define the main pharmacokinetic features of the active ingredients contained within the nutraceutical formulation, i.e. bioavailability, plasma protein binding, blood levels as well as the concentration at the site of action (hair follicles). Obviously, based on the recommended dosage regimen of 800-1000 mg/die, concentrations of 0.5-1 mg/ml such as in vitro are impossible to achieve at the site of action in vivo. This is a common aspect in the development of several drugs and nutraceuticals throughout the transition from in vitro to in vivo models. The concentrations herein used in bioscreen in vitro have been useful both for defining the safety and biocompatibility of the formulation. Using high in vitro concentrations (up to 3 mg/ml in the bioscreen), we have tested the nutraceutical for undesirable cellular responses, thus confirming its safety and complete biocompatibility. As mentioned before, lower concentrations were used to study its effects on the keratin production. However, as emerged by former clinical trials (Tenore et al., J Med Food 2018, 21, 90-103), oral intake of 800 mg in two administration a day of the nutraceutical AnnurtriComplex® for 60 days is able to produce significant effects at the level of the hair follicles. In line with the referee suggestions, on this point we added further information on the recommended dosage in the use of nutraceutical to improve the technical information contained in the manuscript.

Minor points

5. There are many different words for describing dermal papilla cells in the text, such as follicular fibroblasts, dermal cells, dermal papilla, dermal follicular cells, papilla cells. I would like to recommend to unify as “dermal papilla cells” for avoiding confusion because this term is generally accepted in the hair research field.

5. In agreement with the referee we have modified the text by inserting only the definition “dermal papilla cells”.

6. Please paragraph introduction and discussion to improve readability.  

6. Ok, done.

7. Abstract needs to be rewritten because it does not express specific content.

7. We have carefully reviewed the abstract.

8. What is “RFU” described at the right side of the graphs in the Figure 3?

8. RFU stands for Relative Fluorescence Units. We have improved the capture of Figure 3.

Round 2

Reviewer 2 Report

none